# Improving mandibular reconstruction by using topology optimization, patient specific design and additive manufacturing?—A biomechanical comparison against miniplates on human specimen

Jan J. Lang[1,2]☯*, Mirjam Bastian[1]☯, Peter Foehr[1,2], Michael Seebach[3], Jochen Weitz[4], Constantin von Deimling[1,5], Benedikt J. Schwaiger[6], Carina M. Micheler[1,3], Nikolas J. Wilhelm[1], Christian U. Grosse[2], Marco Kesting[7], Rainer Burgkart[1]

1 Department of Orthopedics and Sports Orthopedics, Klinikum rechts der Isar, School of Medicine, Technical University of Munich, Munich, Germany, 2 Department of Mechanical Engineering, Chair of Non-destructive Testing, Technical University of Munich, Munich, Germany, 3 Department of Mechanical Engineering, Institute for Machine Tools and Industrial Management, Technical University of Munich, Munich, Germany, 4 Department of Oral and Maxillofacial Surgery, Josefinum, and Private Practice for Oral and Maxillofacial Surgery at Pferseepark, Augsburg, Germany, 5 Department of Mechanical Engineering, Chair of Applied Mechanics, Technical University of Munich, Munich, Germany, 6 Department of Diagnostic and Interventional Neuroradiology, Klinikum rechts der Isar, School of Medicine, Technical University of Munich, Munich, Germany, 7 Department of Oral and Cranio-Maxillofacial Surgery, School of Medicine, Friedrich-Alexander University Erlangen-Nuernberg, Erlangen, Germany

☯ These authors contributed equally to this work.
* jan.lang@tum.de

**Data Availability Statement:** All relevant data are within the manuscript and its Supporting Information files.

## Abstract

In this study, **top**ology optimized, patient specific **os**teosynthesis plates (TOPOS-implants) are evaluated for the mandibular reconstruction using fibula segments. These shape optimized implants are compared to a standard treatment with miniplates (thickness: 1.0 mm, titanium grade 4) in biomechanical testing using human cadaveric specimen. Mandible and fibula of 21 body donors were used. Geometrical models were created based on automated segmentation of CT-scans of all specimens. All reconstructions, including cutting guides for osteotomy as well as TOPOS-implants, were planned using a custom-made software tool. The TOPOS-implants were produced by electron beam melting (thickness: 1.0 mm, titanium grade 5). The fibula-reconstructed mandibles were tested in static and dynamic testing in a multi-axial test system, which can adapt to the donor anatomy and apply side-specific loads. Static testing was used to confirm mechanical similarity between the reconstruction groups. Force-controlled dynamic testing was performed with a sinusoidal loading between 60 and 240 N (reconstructed side: 30% reduction to consider resected muscles) at 5 Hz for up to $5 \cdot 10^5$ cycles. There was a significant difference between the groups for dynamic testing: All TOPOS-implants stayed intact during all cycles, while miniplate failure occurred after 26.4% of the planned loading ($1.32 \cdot 10^5 \pm 1.46 \cdot 10^5$ cycles). Bone fracture occurred in both groups (miniplates: n = 3, TOPOS-implants: n = 2). A correlation between bone failure and cortical bone thickness in mandible angle as well as the number of bicortical screws used was

**Funding:** This work was supported by the German Research Foundation (DFG) and the Technical University of Munich (TUM) in the framework of the Open Access Publishing Program. The research project "TOPOS - Development, Manufacturing and Testing of Topology Optimized Osteosynthesis Plates" (AZ-1019-12), in whose context the presented study was conducted, is funded by the Bavarian Research Foundation (BFS). The funders had no role in study design, data collection and analysis, decision to publish, or preparation of the manuscript. The funder Josefinum, and private practice for Oral and Maxillofacial Surgery at Pferseepark provided support in the form of salaries for authors JW, but did not have any additional role in the study design, data collection and analysis, decision to publish, or preparation of the manuscript. The specific roles of these authors are articulated in the 'author contributions' section.

**Competing interests:** The authors have declared that no competing interests exist. The affiliation Josefinum, and private practice for Oral and Maxillofacial Surgery at Pferseepark of JW does not alter our adherence to PLOS ONE policies on sharing data and materials.

demonstrated. For both groups no screw failure was detected. In conclusion, the topology optimized, patient specific implants showed superior fatigue properties compared to mini-plates in mandibular reconstruction. Additionally, the patient specific shape comes with intrinsic guiding properties to support the reconstruction process during surgery. This demonstrates that the combination of additive manufacturing and topology optimization can be beneficial for future maxillofacial surgery.

## Introduction

Successful reconstruction of a highly defective mandible is a complex and demanding surgery. Functionality as well as aesthetics are important factors for the outcome and the quality of patients' life. There exist several indications for mandibular reconstruction like resection of tumorous tissue, osteomyelitis, osteonecrosis or trauma [1]. In many cases a free fibular flap is used as substitute for the lost bone [2–5]. This transplantation of a fibular segment includes soft tissue cover and reconnection to the vascular system. For the fixation of the bone material either a reconstruction plate or miniplates are used in most cases.

Reconstruction plates are used as a more rigid alternative to miniplates. However, reconstruction plates have a high volume and adjustment to the mandible curvature is difficult, due to their increased rigidity [6]. Additionally, postoperative plate exposure is often reported with reconstruction plates [7,8]. Miniplates on the other hand come with opposite characteristics. They have a reduced thickness and the surgeon is able to adjust the implants by intraoperative bending to the bone surface. In case of infection single plates can be removed, in contrast to reconstruction plates, which have to be taken out as a whole [9]. Even though, the miniplates come with several advantages, the relatively high failure rate proofs that there is still potential for improvement. It is reported that in around 1 out of 10 reconstructions with miniplates implant failure is detected [2,3]. To increase the routine level and to better support the surgeon in recreating a physiological jawline, virtual planning software is used. Weitz et al. reported that pre-surgical planning significantly improves the outcome of the reconstruction in terms of bone consolidation and reproduction of the native mandibular angle [2]. The common tools for surgical planning need the geometrical data of the patient as input, which is obtained by computed tomography (CT). Consequently, it is a logical step to further exploit this data. Not only the resection of the bone and the positioning of the fibular segments can be planned by using the three-dimensional models of fibula and mandible but also patient individual implants can be created. During planning, virtual implants can be fitted and adapted to the surface of the bone segments. With modern production methods from the field of additive manufacturing of metals for medical application an on-time fabrication of these specific implants can be provided. Using patient specific implants for mandible reconstructions has a positive effect on the treatment process. The guiding properties of these implants lead to a reduced surgery time and costs [10–13]. Additionally, they enable precise reconstructions [14,15] with short duration of postoperative care [16] and fewer complications [10,17].

In this study a new approach for improving the mandibular reconstruction is developed and evaluated. In an interdisciplinary project topology optimized, patient specific implants are created, which combine advantages of miniplates and reconstruction plates. The patient specific implants have a small volume, optimized mechanics and modular exchangeability. Topology optimization is a powerful mathematical tool which allows creating an optimal structural design within prescribed loading and boundary conditions. The material distribution method

for topology optimization is used in a variety of fields like aviation or structural engineering to create specialized construction parts with an optimized design based on an predefined design space [18]. These parts often have a high mechanical stability despite a reduced volume compared to conventional parts. Crudely, this is achieved by iteratively evaluating finite element simulations containing a design space, which has a homogeneous material distribution at the start. After every iteration the material distribution is altered depending on the distribution of the internal stresses in the design space. To obtain the final design, this is continued as long as the boundary conditions (e.g. maximum stress) are not hurt.

For optimization of the evaluated implants in this study, an approach described by Seebach et al. is used [19]. This algorithm creates osteosynthesis plates with a reduced volume while maintaining a high stiffness. In addition, emphasis was placed on an evenly distributed loading of the fixation screws, to prevent stress shielding and screw failure by overloading. The geometry of the implant with the optimized volume leads to a reduced contact area to the bone, which improves the healing capacities due to less periost disturbance. The shape of the implants fits to the bone geometry of the patient and serves the surgeon as guides for the reconstruction during surgery. This can help to reduce the surgery time for mandibular reconstruction. Topology optimization has been described for the design of osteosynthesis plates in maxillofacial surgery in rare cases as virtual design studies [20–22]. But the biomechanical evaluation with biological specimen for these topology optimized implants is still missing in literature.

The aim of this study is a biomechanical evaluation of the of the newly developed implants for mandible reconstruction. This is also used for validation of the production process starting from medical imaging until final reconstruction. Next to the production of the implants with additive manufacturing, this includes virtual planning of the reconstruction as well as automatic design of the implants and cutting guides. For biomechanical evaluation, two different treatment methods for fibula-reconstructed mandibles are compared by static and dynamic testing: standard miniplates versus **top**ology **op**timized, patient specific **os**teosynthesis plates (TOPOS-implants).

## Materials and methods

For this biomechanical study, 21 pairs of fresh frozen (-28°C) cadaveric human mandibles and respective right fibulas were used (Medcure Inc., Portland/OR, USA;). The study was approved by the Technical University ethics committee (607/20 S-KH). The accredited company providing the specimen adheres to AATB standards (American Association of Tissue Banks) and all donors have to provide informed written consent prior to death to be accepted as donors. The average donor age was $67.8 \pm 8.8$ years (mean ± standard deviation). Ten donors were female ($70.0 \pm 10.8$ years) and eleven donors were male ($65.8 \pm 6.3$ years). Considering an equal distribution of sex and age, two groups were created—one for treatment with miniplates (six male, five female; $68.9 \pm 8.1$ years) and one for treatment with TOPOS-implants (five male, five female; $66.6 \pm 9.8$ years). Within these two groups, specimens were split up for static (n = 3; one female, two male) and dynamic (n = 8 resp. n = 7; 4 female, 4 resp. 3 male) testing. All specimens underwent two freezing and thawing cycles during the whole process of preparation, reconstruction and testing. While testing, all specimens were wrapped in moist sheets to avoid dehydration.

High-resolution CT-scans (IQon—Spectral CT) of all frozen specimens were taken using the following standard settings for clinical head scans: tube current intensity 403 mA, tube voltage 120 kV and tube current-time product 300 mAs. Based on the DICOM (Digital Imaging and Communications in Medicine) data of the CT-scans, segmentation was performed

using a Matlab-script (The MathWorks Inc., Natick, Massachusetts, USA) to create geometrical models of all mandibles and fibulae. Thereby, artifacts from implants or other dental treatments were reduced and finally all geometries existed in STL (standard tessellation language) file format. The open-source software *blender* (Blender Foundation, Amsterdam, Netherlands) was used to fix holes in the mesh data and to make sure there was a continuous surface for each 3D model.

A large mandibular LC-defect on the right mandibular side was chosen for the reconstruction. This defect was named according to the classification of Jewer et al. reaching from the right mandibular angle to the left edge of the anterior segment [23]. All individual reconstructions of the mandibles were pre-planned by an experienced surgeon, using a newly developed virtual software tool (Fig 1). This interactive software tool uses the 3D models of the patient's mandible and fibula and enables the surgeon to plan each reconstruction individually, including resection and repositioning. Matching resection guides are created automatically. The recently published software of Raith et al. includes an algorithm that focuses on helping the surgeon to find the best fit of defined fibula segments determined by distinctive parameter setting [24]. In contrast to this, the planning tool used in our study allows a specific adaption of the reconstruction by shifting and rotating all cutting planes and segments freely. After manual placement of three resection planes at the intact mandible, two fibula segments are automatically projected on the resected mandible model. Furthermore, it should be mentioned, that the resection plane in the middle only marks the contact plane of both fibular segments. The mandible does not have to be cut at this position. Manual adaption of the reconstruction can be done freely by movement and rotation of both fibular segments, by changing the position and the angle of the resection planes as well as by altering the harvesting position on the fibula. Doing the virtual reconstruction, the blood supply of the fibula has to be considered because the vessel is supposed to be connected to the corresponding vascular system of the mandible during transplantation for better healing conditions. The performing surgeon did consider this during planning of the reconstruction for this study.

In addition, a data set of specific cutting guides for each reconstruction was created from the geometrical information of the resection planes in STL-format. There was one individual

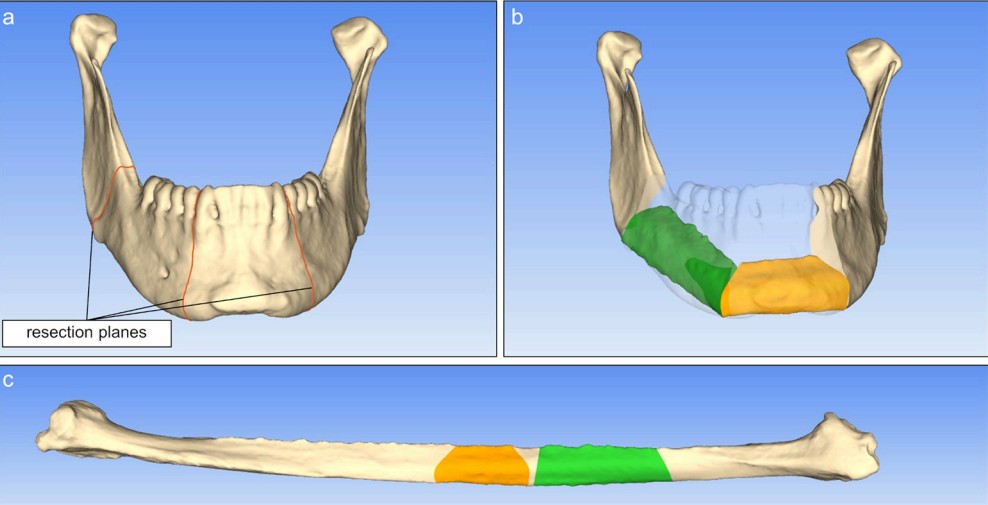

**Fig 1. Virtual reconstruction planning with a custom-made, newly developed software tool.** Patient specific 3D models of mandibula and fibula are used. (a) Resection planes are adjusted freely for the mandible by the surgeon. (c) Fibula segments are automatically projected into (b) the defect and can be adjusted for optimal harvesting site.

cutting guide for each fibula with four saw blade insertion to create the two virtually planned segments. For each mandible two cutting guides were used, one for each jaw angle. All cutting guides were produced via additive manufacturing using standard thermoplastics in fused filament fabrication.

For the newly developed implants, the reconstruction was planned with the use of three master implants, one for each osteosynthesis site of the LC-defect reconstruction. The set of master implants was topology optimized for a LC-defect reconstruction using the geometry of synthetic standard mandible (mandible 8900 from Synbone AG, Malans, Switzerland). The optimization algorithm includes an iterative finite element analysis using Hyperworks and OptiStruct (Altair Engineering, Troy, USA) to achieve an implant with high stiffness despite having a reduced volume to 45% on average (compared to design space). In other words, the optimization is a material reduction process that results in the least possible loss of stiffness for the implant. Another aim for the optimization was an even stress distribution among the bone screws to prevent screw failure. This optimization process was described before by Seebach et al [19]. For the presented study the shape of these topology optimized master implants was fitted to ten virtual reconstructions of the TOPOS group based on the CT data of the specimens (Fig 2). With this consecutive optimization of topology and shape, 30 patient specific plates (three per reconstruction) with a thickness of 1 mm were created and produced by electron beam melting (FIT Production GmbH, Lupburg, Germany) using medical-grade titanium alloy (Ti6Al4V, ISO 5832-3/ASTM F1472) as described by Seebach et al. [25]. For post-processing, grinding and polishing were applied.

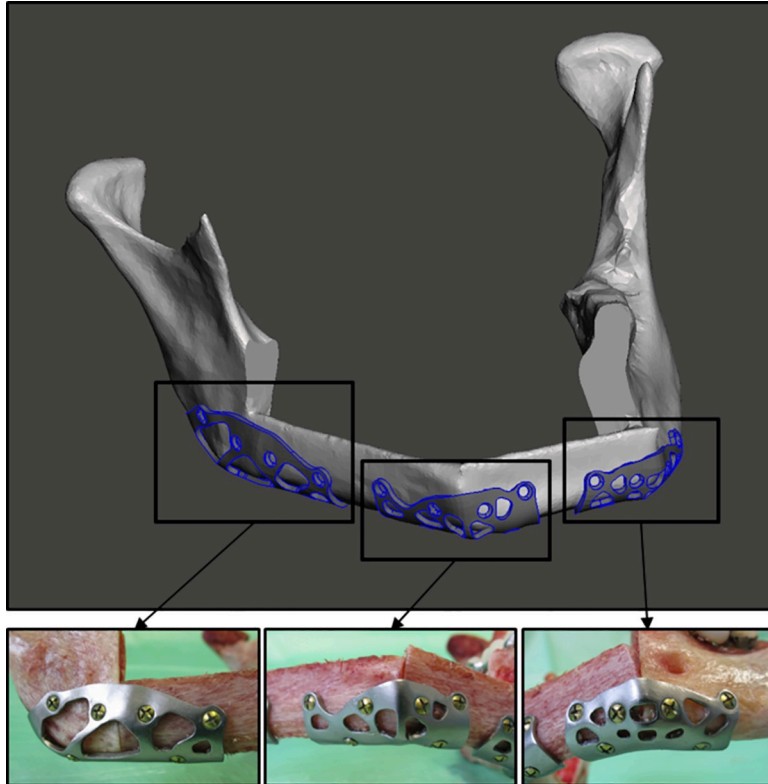

**Fig 2. Patient specific, topology optimized osteosynthesis plates (TOPOS-implants).** (Top) Virtual planning and fitting of the osteosynthesis plates for the fixation of the fibular segments. (Bottom) Application of the implants to the specimen produced with electron beam melting.

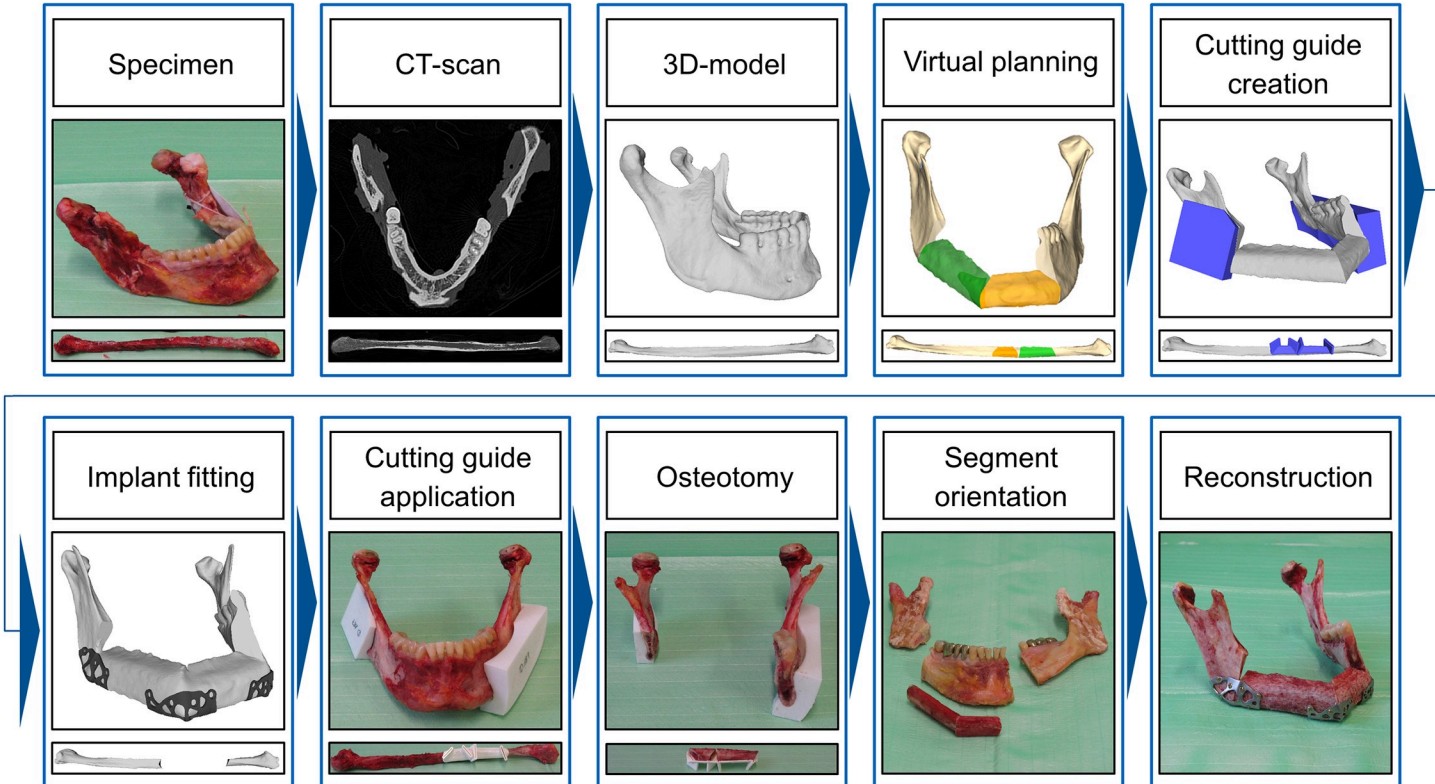

**Fig 3. Process from bone to reconstruction.** Stepwise process of creating the reconstruction specimen with topology optimized patient specific implants from the intact bone to the reconstructed mandible for testing. An experienced surgeon performed virtual planning and reconstruction. For reconstruction with miniplates the steps are similar but without the implant fitting.

The preparation of all specimens included removing of soft tissue including muscle attachments, blood vessels, fat and periosteum. This was done for the cutting guides to fit accurately because they are created on the base of CT-data and soft tissue was not included in the segmentation. For osteotomy the specific cutting guides were applied on the mandible and fibula (Fig 3). During the reconstruction of a mandible, the first step was to arrange and combine both fibular segments. Afterwards they were connected with the corresponding mandible segment on each side.

In case of the reconstruction with miniplates (cranial plate, straight, rigid; plate thickness 1.0 mm; material: titanium grade 4 (ASTM F67); Modus 2.0, Medartis AG, Basel), the implants for joining fibular and mandibular segments had to be aligned to the individual bone geometry by an experienced oral and maxillofacial surgeon. Plates with four holes (bar: 9 mm) and plates with six holes (bar: 9 mm or 12 mm) as well as appropriate screws (cross; material: titanium grade 4 (ASTM F67); Modus 2.0, Medartis AG, Basel) with a length 6 mm or 9 mm were used. The fixation of corresponding bone segments was performed with standard surgical equipment. The position of the screws was chosen for every reconstruction individual by surgeon. The TOPOS implants were fixated to the bone with the same kind of screws as used for the miniplates. For all reconstructed mandibles, a standardized hole (d = 5 mm) at the middle of each ramus mandibulae near the foramen mandibulae was drilled for fixation of the specimens in the test system and the processus coronoideus was removed.

After reconstruction of the mandibles either using miniplates or TOPOS-implants, static and dynamic testing was performed on a custom-made multi-axial test system. The system

allows adapting to the anatomical conditions of the donor and applying side specific loads [26]. For the determination of the applicable load direction and appropriate bearings for the test setup, a model with all relevant muscles that are part of chewing was developed. The orientation of the muscle forces of the major muscle groups were taken from literature [27–29]. The combination of the muscle orientations for the mandible was used to derive the direction of the force for testing on each side of the mandible. The resulting direction of the force, which was applied in the test system, is mainly orthogonal to the chewing plane with a pivoting of 7˚ anterior. Tilting of the force direction of calculated 0.4˚ to lateral was neglected for the setup. Three bearing points were used for the model and derived to the testing setup. The bite point was defined as the anterior end of the left segment (first premolar if present), since the incisors are missing for the reconstructions. Choosing this bearing point position and the front edge of the mandibular segment created high bending moments for the reconstruction. This bite point was only constrained in vertical direction [30]. This allows frictional displacement in the horizontal plane for the bite point. The capita mandibulae were the remaining two bearings, which were constrained in two translatory directions. Allowing a limited, frictional displacement in medio-lateral direction is in contrast to the conditions in numerical models [29,31,32]. But the risk of asymmetrical load distribution due to asymmetrical bearing is reduced.

The custom-made test system, which allowed mechanical testing of mandibular reconstructions to analyze the fatigue behavior of the osteosynthesis plates according to DIN 50100, was force-controlled and it was possible to test two specimens at the same time. Load was applied through two bowden cables per mandible (one for each side, Fig 4), that followed the orientation of the resulting force vector and lead to hydraulic pistons via deflection rollers. For each of the cables a load cell was incorporated into the loading line, which was used for force control and documentation. Aluminum bolts with semicircular grooves at the lower end were used to support the mandibular condyles. These bearings were adjustable to the geometry of the specimens by altering the height and orientation of the groove position. At the opposite side, there was a metallic angle bracket that simulates the maxilla and which could be moved in three directions to adjust the position. That means the left front part of the mandible was in contact with the angle so movement in vertical direction was inhibited at this point (Fig 4), similar to the setup of Schupp et al. [33].

To analyze the deformation and to gain information about stiffness and maximum load up to failure during static testing, two linear potentiometric position sensors (MMR10-11, Megatron, Germany; range: 10 mm) connected to a 16-bit-analog-digital converter were attached. Sensor 1 was used to measure displacement in vertical direction at the contact point of the fibular segments while sensor 2 registered lateral displacement on the right mandibular angle (Fig 4C and 4D). At the beginning of each test run, both sides of the mandible were preloaded with 20 N.

For static testing (n = 3 per implant group; one female, two male), a force ramp was applied to the mandibular angles up to 500 N. Failure of the bone-implant-construct was determined as fracture or obvious deformation of bone or implant. Sensor 1 and sensor 2 were measuring the vertical and lateral displacement of the reconstructed mandible. The end of the linear elastic region (elastic limit) was determined by considering the load-displacement-diagrams of both sensors. The results from the static testing were used to set the maximum load for the dynamic testing of the reconstructions and to validate the mechanical similarity of the reconstruction methods.

The maximum force for dynamic testing $F_{l,max}$ was defined as 85% of the force at elastic limit, which was derived from the displacement measurements during static testing. The maximum load on the reconstructed side of the mandible $F_{r,max}$ was reduced by 30%, in order to consider the resection of muscles during surgery and reduced chewing loads, similar to

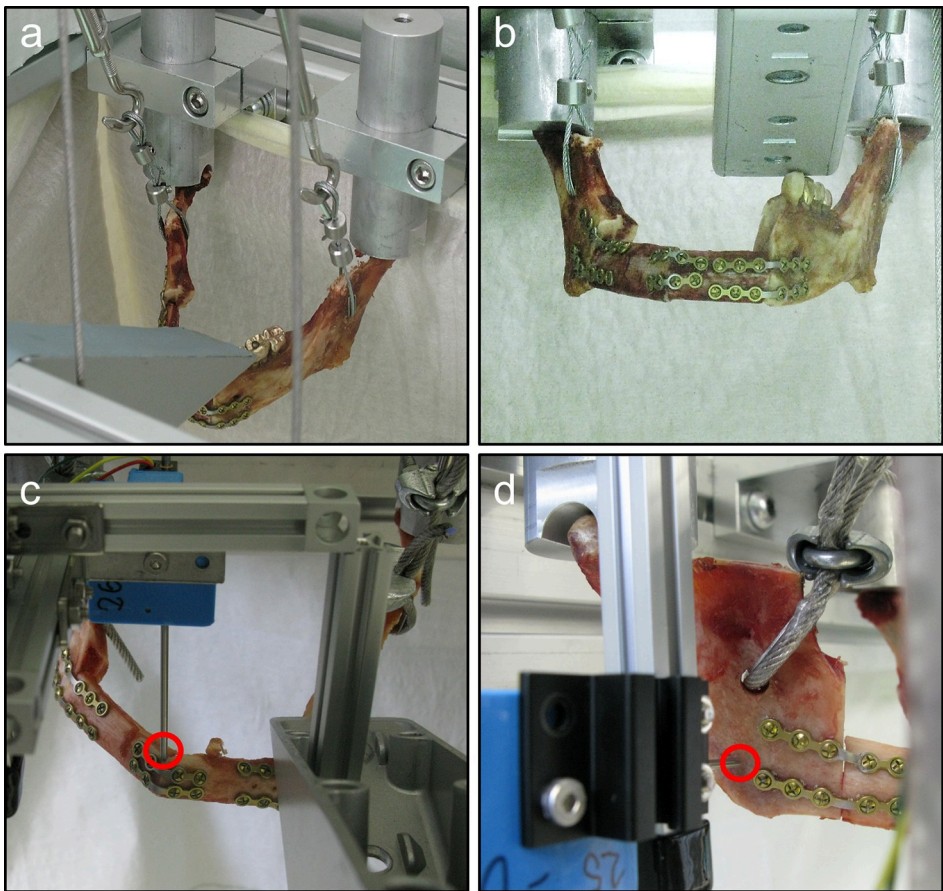

**Fig 4. Biomechanical test setup for dynamic and static testing.** (a) test setup for dynamic testing of the constructed specimen, force is applied through the cables fixed to the ramen mandibulae (b) front view of the dynamic test setup with an exemplary reconstruction using miniplates (c and d) static testing of the reconstruction in the same test rig as for dynamic testing with additional displacement sensors measuring (c) the vertical displacement on the mandible (red circle) at the contact plane of the fibula segments and (d) the lateral displacement at the ramen mandibulae.

the approach of Schupp at al. [33]. Sinusoidal loading between $F_{min}$ = 60 N and $F_{l,max}$ = 240 N ($F_{r,max}$ = 168 N) at a frequency of 5 Hz was performed for up to $5 \cdot 10^5$ cycles to analyze the fatigue properties of both treatment methods (miniplates: n = 8; TOPOS-implants: n = 7). The frequency was chosen to reduce the effect of biological degradation as far as possible. The maximum piston speed for load changes at 5 Hz was sufficient to provide the expected displacements for the applied load interval. After $N_E$ = 100 cycles all subsidence processes are finished and stable testing can be supposed. For all following cycles, a change in the maximum displacement of more than 5 mm, related to the position at $N_E$, was considered as failure of the reconstructed mandible and led to an automatic stopping.

To validate the variation in bone quality from the different donors the thickness of the cortical bone was measured at the ramen mandibulae according to Heibel et al. with a caliber [34]. Three measurement sites were taken at the resection plane of the right mandibular angle for evaluation (Fig 5).

Since the distribution of the results in the different groups deviated from a normal distribution, which was tested with a Shapiro-Wilk normality test, the conditions for a standardized Student's t-test were not met. Due to that, a nonparametric Mann-Whitney U test was used to analyze the statistical significance of the difference between the groups. Standard significance

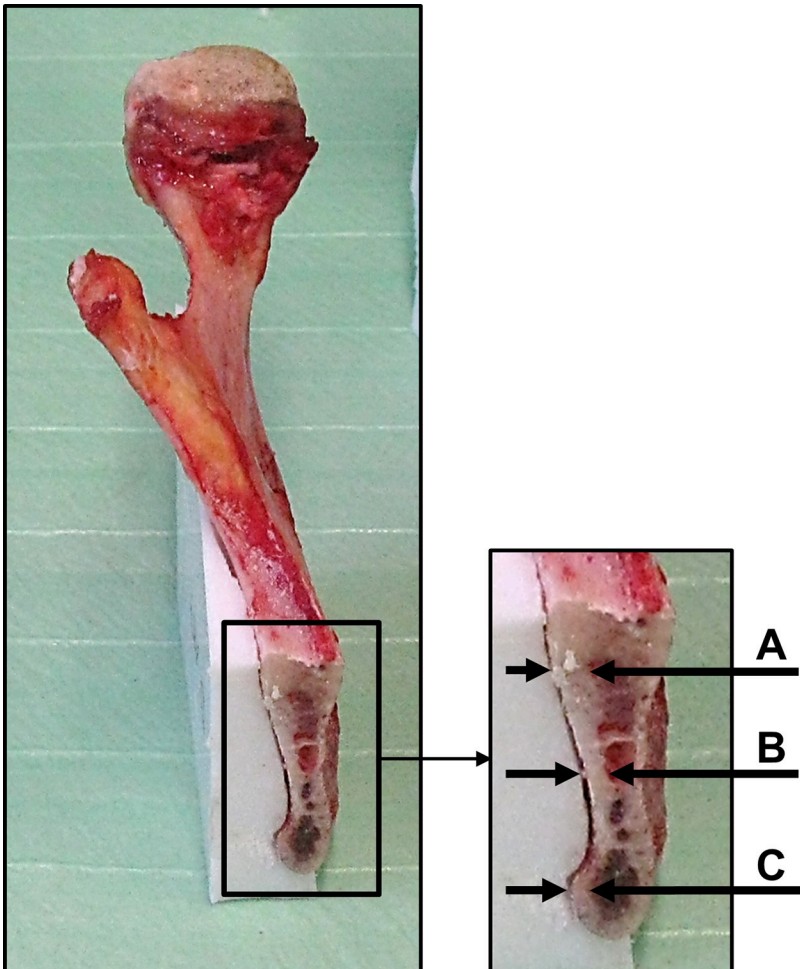

**Fig 5. Measurement of the cortical thickness of the mandible at the resection plane of the right mandibular angle.** Measurement of the cortical thickness of the mandible according to Heibel et al. [34].

thresholds of $p < 0.05$ and $p < 0.01$ were used. The evaluation was performed with the software tool Prism 6.07 (GraphPad Software, Inc., San Diego, USA).

## Results

In static testing, the elastic limit of bone-implant-constructs for the group with miniplates was reached at 287 ± 57 N (mean ± standard deviation) while the TOPOS-implant group had their mean value at 276 ± 125 N. In the elastic deformation range a vertical stiffness of 181 ± 33 N/mm was derived for the reconstruction with miniplates and 151 ± 16 N/mm for reconstruction with TOPOS-implants. At the time of failure, a vertical displacement of 2.6 ± 1.3 mm was measured for reconstructions with miniplates and 2.5 ± 1.4 mm for reconstructions with TOPOS-implants. Lateral displacement of the right mandibular angle was smaller for specimens treated with miniplates (0.7 ± 0.3 mm) than for those with TOPOS-implants (2.0 ± 1.6 mm). Failure of the reconstructions occurred at 413 ± 115 N (miniplates) and 400 ± 217 N (TOPOS-implants). Occurring failure types were bone or tooth failure and massive plastic deformation of the implant. This deformation of the implant happened only to one reconstruction with miniplates (S1 Table).

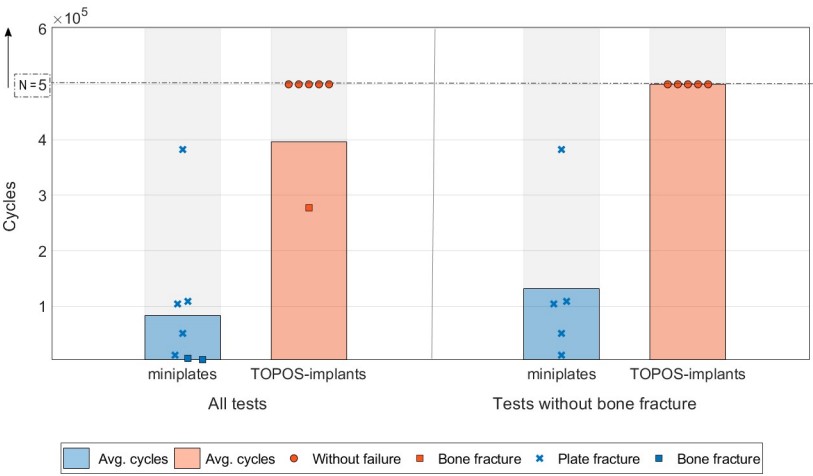

**Fig 6. Results of the dynamic tests.** Graphical visualization of the biomechanical comparison of miniplates with patient specific, topology optimized (TOPOS-)implants. The bars show the average of the passed cycles per group with 500 000 cycles at maximum. For comparing of fatigue properties of the implant types only test runs without bone fracture are used (statistics: significant ($^*$) p<0.05, highly significant ($^{**}$) p<0.01).

Even though the reconstructions of both groups showed a very similar mechanical behavior under static loading, in cyclic testing a significant difference between the groups was observed. A distinction was made between three different outcome types: bone fractures, plate fractures and specimens that passed all applied cycles without failure (Fig 6). Screw failure was not observed for all tests. Passing all applied cycles without failure only occurred within the group of TOPOS-implants, where five reconstructions passed $5 \cdot 10^5$ cycles without failures. Plate fractures solely happened in the miniplate group in five out of eight cases (Fig 7C). Three of these fractures were localized at a plate that connected the right mandibular segment with the lateral fibular segment. Two plate fractures occurred at a miniplate holding on to the left mandibular segment and the central fibular segment. Bone fractures at the right ramus mandibulae appeared in both groups (for miniplates: n = 3, for TOPOS-implants: n = 2). In these cases the fracture line was always running through the upper screw holes in the bone for the corresponding implant (Fig 7A and 7B) When disregarding bone fractures, it is shown that reconstructions with miniplates reached an average number of $1.32 \cdot 10^5 \pm 1.46 \cdot 10^5$ cycles (n = 5). A highly significant difference between the groups can be determined because no TOPOS-implant failed for the applied cycles (Mann-Whitney U = 0; p = 0.008<0,01 two-tailed). Considering all specimens, reconstruction failure (including bone fracture) is noted after $3.97 \cdot 10^5 \pm 1.94 \cdot 10^5$ load cycles in the TOPOS-implant group and at $8.37 \cdot 10^4 \pm 1.29 \cdot 10^5$ load cycles after treatment with miniplates. Statistically this also corresponds to a significant difference between the groups (Mann-Whitney U = 8; p = 0.026<0,05 two-tailed). Statistical analysis was performed according to two non-normally distributed groups (Shapiro-Wilk: miniplates group p = 0.002; TOPOS-group p = 0.0008).

The results for the measuring of the cortical thickness over all specimen are shown in Table 1. The measurement was performed at three sites, while b corresponds to the site were all fractures of the ramen mandibulae occurred.

## Discussion

The biomechanical testing in this study was performed on an in-house developed testing setup. Using a custom-made solution like this comes with the disadvantage of limited

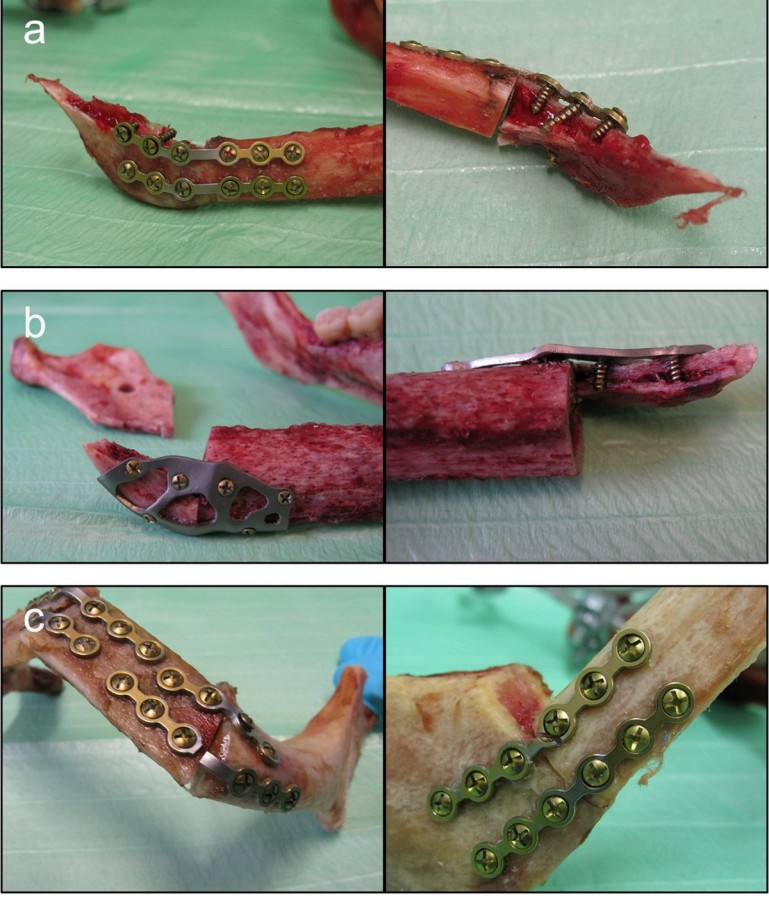

**Fig 7. Failure cases of the biomechanical testing.** Bone fracture occurred at the right mandibular angle with the fracture line running across the upper fixation screws for (a) miniplates (two views) and (b) TOPOS-implants (two views); (c)Failure of the implant only occurred for reconstructions with miniplates at both fixation sites of the mandibular segment to the fibular segment(left: left mandibular segment to central fibular segment; right: right mandibular segment to right fibular segment).

reproducibility for third parties. But a commercially available solution does not exist for this testing scenario. And due to the detailed description of the testing and the publication presenting the experimental setup by Foehr et al. [26], distinct transparency, comprehensibility and adaptability of the testing conditions are ensured.

In static testing the reconstructions with the TOPOS-implants as well as miniplates behaved quite similar. Elastic limit and maximum force until failure differ only 4.2% and 3.1% respectively. Furthermore, fractures occur at the same site and the vertical displacement is almost the

**Table 1. Measurement results of the cortical thickness of the mandible for all specimen (mean ± standard deviation).**

|  | A | B | C |
|---|---|---|---|
| All specimen | 2.2 ± 0.6 mm | 1.5 ± 0.5 mm | 1.9 ± 0.5 mm |
| Specimen with bone fracture | 1.6 ± 0.3 mm | 1.0 ± 0.2 mm | 1.7 ± 0.4 mm |
| Specimen without bone fracture | 2.7 ± 0.3 mm | 2.0 ± 0.3 mm | 2.2 ± 0.5 mm |

Measurement locations A, B and C can be seen in Fig 5.

same. However, the lateral displacement is larger for reconstructions with TOPOS-implants. We assume, that the shape of the implant, which is fitted to the bone geometry, creates a more rigid structure of all four bony segments. This enables an increased force transmission between the segments that are close to each other. As a consequence, the left side with the higher load would have more influence on the reconstructed right side, at which the displacement had been measured. Consequently, the local loading and thus the displacement at the right mandibular angle would be higher. However, the measured difference in the average displacements of 1.33 mm also can be seen as clinically uncritical. Furthermore, a group size of n = 3 is too small to make reliable statements on this. Which is why the standard deviation in both groups for the maximum load is relatively high. In both groups the female mandibles, which were noticeably smaller, failed distinctly earlier (280 N (miniplates) resp. 150 N (TOPOS-implants); n = 1 for each group) compared to the male specimens (at 480 ± 14 N (miniplates) resp. 525 ± 21 N (TOPOS-implants); n = 2 for each group). This caused a high deviation regarding the failure load. The difference in the determined stiffness for the reconstruction types in static testing suggests that a slightly stiffer reconstruction of the mandible can be achieved with miniplates. But the actuator displacement during cyclic testing, which is range that the actuator moves from minimum load to the maximum load in one cycle, cannot confirm that (S2 Table). This displacement, which is also a measure for the stiffness of the reconstruction, is very similar for both reconstruction groups. The mean actuator displacement for the reconstruction with TOPOS implants was 2.1 ± 0.3 mm for the right side and 2.7 ± 0,24 mm for the left side at cycle 100. For the reconstruction with miniplates an actuator displacement of 2.2 ± 0.3 mm on the right side and 2.9 ± 0.3 mm on the left side was determined at the same cycle. This underlines the mechanical comparability of the two reconstruction approaches. The mechanical similarity also confirms that there is no excessive stiffening caused by the TOPOS implants. If the bone fragments are positioned too rigidly for osteosynthesis, it can result in inhibition of sufficient micromotions in the fracture gap [35,36]. Stress shielding like effects can lead to impaired healing in this context due to low interfragmentary strains at the fracture site [37]. But as already described there is no indication for this found in mechanical testing.

The mechanical behavior of both specimen groups shows to be quite similar in static testing, but for the dynamic testing, the differences are more distinctive. The cyclic testing demonstrated clearly, that the TOPOS-implants, which had been optimized considering maximum stiffness and evenly distributed stresses, showed significantly better fatigue properties compared to miniplates. This is still true when considering only specimens without bone fracture. None of the TOPOS-implants failed during the $5 \cdot 10^5$ cycles, while the miniplates only reached 26% of these cycles on average. It is likely that the reconstructions with TOPOS-implants would have lasted significantly more cycles. Which means stopping the dynamic loading after half a million cycles puts the miniplates in a favourable light. The difference is therefore even more distinctive than shown here.

Considering the diversity in shape and geometry of human bones, it also becomes apparent in Fig 6, that there is a huge variation in proportions of mandibles and corresponding fibulae. While in the Fig 6A mandible and fibula of the bone combination are approximately of the same width, the combination in Fig 6B shows a mandible that is in total (at the pictured resection plane) only as wide as the cortical shell of the corresponding fibula. However, only combinations of the same donor were used.

Overall, nine out of 21 specimens failed because of bone fracture at the right ramus mandibulae. The fracture line ran along the line of the drilled holes for the upper miniplate respectively the upper holes of the corresponding TOPOS-implant. Consequently, it seems to be obvious that there is a link between the weakening of the bone structure through screws and

the failure of the reconstructed mandible. It could be observed that in case of a bone fracture the performing surgeon used more often 9 mm screws instead of 6 mm at the right mandibular segment for fixation of the implants. This was done subjectively as compensation of structural instability. The longer screws enabled a bicortical fixation, while the 6 mm screws functioned monocortical. Considering the whole number of screws at the right mandible (six screws for miniplates, three screws for TOPOS-implants), 42% of screws were the long version in case of bone fracture, whereas only 16% were the long version when there was no bone fracture. Nevertheless, the question arises if either the usage of many long screws weakens the bone or if on an already weak bone with a high risk of fracture more 9 mm screws are used. According to Heibel et al., the corresponding region of the ramus mandibulae is the area with lowest thickness of the cortical bone in the mandible [34]. Also, 23–42% of all clinically seen mandible fractures happen at the mandibular angle, which underlines the natural mechanical instability of this area [20,38]. Additionally, the specimens with bone fracture showed a lower mean value of cortical bone thickness (1.0 ± 0.1 mm) than the average over all mandibles in the corresponding area (1.5 ± 0.5 mm) as well as the specimens, that did not fail because of bone fracture (2.0 ± 0.4 mm) (Table 1B). Due to this the area at the ramen mandibulae seems to be especially prone to fracture. Consequently, fixation in this area should be avoided or the number of used screws should be reduced for clinical mandibular reconstruction in cases of poor bone quality.

The approach of using patient specific osteosynthesis plates to improve mandibular reconstruction is not new. Since 2013 a patient individual concept is provided by DePuy Synthes (DePuy Synthes, J&J Medical Devices, Warsaw, USA) in cooperation with Materialize (Materialize, Leuven, Belgium) which supports the presurgical planning of reconstruction plates to the needs of the surgeon [39,40]. Rendenbach et al. used a similar defect model as presented here in synthetic mandibles and compared a CAD/CAM designed reconstruction plate to treatment with miniplates and conventional reconstruction plates [40]. The standard reconstructions showed inferior mechanical properties in fatigue and stiffness compared to the CAD/CAM design. Also other companies like Stryker (Stryker, Kalamazoo, USA) and KLS Martin Group (Gebrüder Martin GmbH & Co. KG, Tuttlingen, Germany) are commercially providing patient individual implant planning and production [41,42]. Gutwald et al. even performed an optimization of the patient specific reconstruction plate by using a sensitivity analysis of 72 predefined options of the implant [42]. But the ultimate goal for the design of osteosynthesis plates should be that they provide stability to ideally support the healing of the bone, while interfering with it as little as possible. Topology optimization is a suitable option to reach this goal. Our study shows the first experimental evaluation of a topology optimization approach on osteosynthesis plates for reconstruction, but topology optimization has been seen in maxillofacial surgery before. Lovald et al. used topology optimization for creation of osteosynthesis plates for fracture treatment at the symphysis and lateral body of the mandible [21,22]. These optimized implants showed better results in plate stresses and fracture strain in comparison with standard treatments including miniplates using finite element analysis. Liu et al. also used topology optimization to design a V-shaped implant for the treatment of mandibular angle fracture [20]. The plate was numerically evaluated against standard osteosynthesis with miniplates of the same thickness. The simulations with the topology optimized implants had lower von-Mises stresses on the implant as well as a reduced fracture displacement. Finite element analysis is an important tool in the field of biomechanics, which is widely used to analyse the complex biomechanical behavior of the mandible. But disadvantages come with an evaluation of the optimized implants with finite element analysis. Simplifying the screw geometry with a cylinder leads to a distinct underestimation of the peak stresses in screws [43]. Furthermore, the reproduction of the complex reality of bone tissue is difficult in numerical models. In most

simulations bone is divided in cortical and cancellous bone with isotropic material parameters each. In more sophisticated approaches the material parameters are dependent on the mineral content of the bone using CT data. But in reality, bone is a highly complex structure with anisotropic and viscoelastic material properties, which are not considered in numerical models. Also, an evaluation of the optimized implant on the same model and loading condition as it was created on is only the first step in the validation process of the implant. Comparing it numerically to standard implants that can be used universally, validates the topology optimization algorithm rather than the functional performance of the implant. Due to this we evaluated our topology optimized, patient specific implants on cadaveric models. This makes an application of the physiological loading conditions more difficult but provides a better representation of the material properties and the diversity of bone quality and geometry. Furthermore, we used a practical approach that puts the topology optimization before the adapting of the implants to the patient. This generalized optimization is very helpful in increasing the speed of generating an implant for different patients, which makes the integration of topology optimization into a fast and flexible treatment process much easier.

For the evaluation of the whole process from the planning with the software to the production of the implants, reconstruction of the mandibles and finally biomechanical testing, the analysis was focused on LC-defects. But the concept of topology optimization and fitting of the shape to the patient's bone geometry can be adapted for all kind of application of osteosynthesis plates.

## Conclusions

This study displays a significantly increased cyclic loading capacity of mandible reconstructions with topology optimized and patient specific implants in comparison with miniplates as clinical standard. Due to their shape fitted to the bone surface they also function as guides for the surgeon to create the planned reconstruction, which can reduce surgery time. The presented combination of generalized optimization and consecutive shape adapting to the patient specific surface creates a time-efficient possibility to create optimized, patient specific implants. Additionally, the introduced implants share the advantages of miniplates by being site-specific exchangeable and not protruding as much as reconstruction plates. As additive manufacturing techniques are becoming more and more established in medical applications, the production costs will fall. Consequently, patient specific osteosynthesis plates, which are optimized to increase stability, could become widely used in mandibular reconstruction in future.

## Supporting information

**S1 Table. Measurement results of static testing and the cortical thickness of the mandible.**
(DOCX)

**S2 Table. Measurement results dynamic testing and the cortical thickness of the mandible.**
(DOCX)

## Acknowledgments

FIT Production GmbH (Lupburg, Germany) produced the additively manufactured, patient-specific bone plates used in this study.

## Author Contributions

**Conceptualization:** Peter Foehr, Jochen Weitz, Marco Kesting, Rainer Burgkart.

**Data curation:** Jan J. Lang.

**Formal analysis:** Jan J. Lang, Mirjam Bastian.

**Funding acquisition:** Peter Foehr, Michael Seebach, Marco Kesting, Rainer Burgkart.

**Investigation:** Jan J. Lang, Mirjam Bastian, Peter Foehr, Jochen Weitz, Benedikt J. Schwaiger.

**Methodology:** Jan J. Lang, Peter Foehr, Jochen Weitz, Constantin von Deimling, Carina M. Micheler, Nikolas J. Wilhelm, Rainer Burgkart.

**Project administration:** Jan J. Lang, Peter Foehr, Michael Seebach, Benedikt J. Schwaiger, Rainer Burgkart.

**Resources:** Carina M. Micheler, Nikolas J. Wilhelm, Christian U. Grosse, Marco Kesting, Rainer Burgkart.

**Software:** Constantin von Deimling.

**Supervision:** Christian U. Grosse.

**Validation:** Jan J. Lang, Benedikt J. Schwaiger, Carina M. Micheler, Nikolas J. Wilhelm, Rainer Burgkart.

**Visualization:** Jan J. Lang.

**Writing – original draft:** Jan J. Lang, Mirjam Bastian.

**Writing – review & editing:** Jan J. Lang, Mirjam Bastian, Peter Foehr, Michael Seebach, Jochen Weitz, Constantin von Deimling, Benedikt J. Schwaiger, Carina M. Micheler, Nikolas J. Wilhelm, Christian U. Grosse, Marco Kesting, Rainer Burgkart.

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
