## [Decision Letter · Decision Letter 0]

21 Jan 2021

PONE-D-20-39788

Improving mandibular reconstruction by using topology optimization, patient specific design and additive manufacturing? – A biomechanical comparison against miniplates on human specimen

PLOS ONE

Dear Dr. Lang,

Thank you for submitting your manuscript to PLOS ONE. After careful consideration, we feel that it has merit but does not fully meet PLOS ONE’s publication criteria as it currently stands. Therefore, we invite you to submit a revised version of the manuscript that addresses the points raised during the review process.

The manuscript is a well-conducted study providing novel findings in a relatively new field. Please address the comments/suggestions from the academic editor and reviewer 1.

We look forward to receiving your revised manuscript.

Kind regards,

Luis Cordova

Academic Editor

PLOS ONE

Journal Requirements:

2. In the ethics statement in the manuscript and in the online submission form, please provide additional information about the human tissues used in this study. Specifically, please ensure that you have discussed whether next-of-kin provided informed written consent for the use of the tissues. If patients provided informed written consent prior to death to have their bodies used in medical research, please include this information.

3. Please list the name and version of any software package used for statistical analysis, alongside any relevant references. For more information on PLOS ONE's expectations for statistical reporting, please see https://journals.plos.org/plosone/s/submission-guidelines.#loc-statistical-reporting.

4. Please ensure you have thoroughly discussed any potential limitations of this study within the Discussion section, including the potential impact of confounding factors.

6.Thank you for stating the following in the **Financial Disclosure** section:

"This work was supported by the German Research Foundation (DFG) and the Technical University of Munich (TUM) in the framework of the Open Access Publishing Program.

The research project “TOPOS - Development, Manufacturing and Testing of Topology Optimized Osteosynthesis Plates” (AZ-1019-12), in whose context the presented study was conducted, is funded by the Bavarian Research Foundation (BFS).

We note that one or more of the authors are employed by a commercial company: Josefinum, and private practice for Oral and Maxillofacial Surgery at Pferseepark

Additional Editor Comments:

The authors propose a well-conducted study to compare topology optimized, patient specific osteosynthesis plates (TOPOS-implants) versus 1.0 mm miniplates by biomechanical testing using fibula-reconstructed cadaveric mandibles.

Introduction: To increase the clinical relevance, please include data about the benefits of PSI versus plates, referring to some clinical outcomes: recovery time for patients, shorten surgical and hospital time, etc.

The aim of the study should be stated

The topology optimization of implants, the new technique tested in this manuscript, is defined as “…a powerful mathematical tool which allows creating an optimal structural design within prescribed loading and boundary conditions.”. This definition was supported by just one reference (#10) in page 4 line 76. This is a key point for this manuscript, so please add a more detailed definition and features of this specific technique to lead medical readers.

Conclusions: They should state more precisely the benefits of TOPOS-implants over miniplates rather than “significant superiority”, which is an unprecise and subjective word.

Reviewers' comments:

Reviewer's Responses to Questions

**Comments to the Author**

1. Is the manuscript technically sound, and do the data support the conclusions?

Reviewer #1: Yes

Reviewer #2: Yes

2. Has the statistical analysis been performed appropriately and rigorously? 

Reviewer #1: Yes

Reviewer #2: Yes

3. Have the authors made all data underlying the findings in their manuscript fully available?

Reviewer #1: No

Reviewer #2: Yes

4. Is the manuscript presented in an intelligible fashion and written in standard English?

Reviewer #1: Yes

Reviewer #2: Yes

5. Review Comments to the Author

Reviewer #1: Abstract

Line 26: Please mention that “1.0mm” refers to thickness. Define “miniplates” and disclose their titanium grade.

Line 29: Thickness and other dimensions of the TOPOS plates should be mentioned.

Line 30: Titanium alloy? Grade?

In Lines 212-215 you explain that the TOPOS plates fractured under lower static load than the miniplates. Also that the fibula fractured more quickly, but SD probably did not allow to reach statistical significance. Should this not be mentioned?

The shape of the implant seems more important than any other characteristics (line 279). This can be mentioned in the abstract.

In general, the abstract can reflect the findings in a much better and much understandable way.

M&M

Line 135: is high stiffness (= high brittleness) correctly the goal? Would reduced plasticity and increased elasticity not be a better criterion? Please explain in your text.

Line 164: what do you mean with “surgical procedures”?

Line 165: why did you choose for the same (small diameter) screws? Luckily none failed. Is the position of the screws not important? The configuration should be explained and the influence of the latter on bone fracture.

Line 176-182: this is difficult to understand. Please rephrase. Did you take laterotrusion (= horizontal shear forces at the occlusal level in the molar area) and protrusion (= vertical shear forces at the incisors) into account?

Line 180: name of device, name of company, city, country

Lines 212-218 belong to the Results section.

Line 230: could Young modulus not be determined per specimen after static and cyclic loading, to test differences between mandibles, being probably a confounding factor?

Line 236: did you perform a Shapiro-Wilk test to conclude this?

Results

Line 253: I read “No TOPOS-implant failed for the applied cycles” and then I read in Line 254: “Failure is noted in the TOPOS implant group”, which is confusing.

Overall a good paper, but difficult to read for a surgeon. Its impact may increase of you make it more accessible.

Reviewer #2: According to the study design by the authors, the research work it is according to be in the right way by the literature available . Could be interesting in the future to use the same design of study to evaluate the properties of PSI vs reconstruction plates. The only thing that I miss was the blind selection of the groups of miniplates vs PSI.

6. PLOS authors have the option to publish the peer review history of their article (what does this mean?). If published, this will include your full peer review and any attached files.

Reviewer #1: **Yes: **Maurice Yves Mommaerts

Reviewer #2: **Yes: **Rolando Carrasco

---

## [Author Response · Author response to Decision Letter 0]

16 Apr 2021

In agreement with the PLOS ONE style templates the section names were removed from the first page.

2. In the ethics statement in the manuscript and in the online submission form, please provide additional information about the human tissues used in this study. Specifically, please ensure that you have discussed whether next-of-kin provided informed written consent for the use of the tissues. If patients provided informed written consent prior to death to have their bodies used in medical research, please include this information.

Additional information about company providing the specimen and the informed written consent was included in the online submission form and the manuscript. 

3. Please list the name and version of any software package used for statistical analysis, alongside any relevant references. For more information on PLOS ONE's expectations for statistical reporting, please see https://journals.plos.org/plosone/s/submission-guidelines.#loc-statistical-reporting.

The name and version of the software for statistical analysis was added to the manuscript. 

4. Please ensure you have thoroughly discussed any potential limitations of this study within the Discussion section, including the potential impact of confounding factors.

Several factors and their influence on the results are identified and discussed in this study. The parameters sex and age were taken into account by distributing them evenly among the implant groups. Bone quality is another limiting factor that comes together with individual biological specimen. It is shown that the quality of the bone varied greatly. In addition, it was investigated whether the mechanical stability, in particular the stiffness, of the reconstructions differed between the implant groups. This was highlighted in more detail in an additional section of the manuscript. No relevant difference between the stiffnesses of the reconstructions could be found in the static testing as well as in the dynamic testing. Thus, this parameter can be excluded as a confounding factor. In addition, for the reconstruction of the mandible, an area on the mandible was identified that is easily weakened by the application of screws. Bone fracture in this area is a prominent failure type in this study. It indicates that for further development of the implant, avoiding the fixation in this area could be beneficial for stability in case of low bone quality.

The phrase referring to these data was removed from the manuscript. 

6.Thank you for stating the following in the Financial Disclosure section:

"This work was supported by the German Research Foundation (DFG) and the Technical University of Munich (TUM) in the framework of the Open Access Publishing Program.

The research project “TOPOS - Development, Manufacturing and Testing of Topology Optimized Osteosynthesis Plates” (AZ-1019-12), in whose context the presented study was conducted, is funded by the Bavarian Research Foundation (BFS).

We note that one or more of the authors are employed by a commercial company: Josefinum, and private practice for Oral and Maxillofacial Surgery at Pferseepark

Updated Funding Statement:

This work was supported by the German Research Foundation (DFG) and the Technical University of Munich (TUM) in the framework of the Open Access Publishing Program.

The research project “TOPOS - Development, Manufacturing and Testing of Topology Optimized Osteosynthesis Plates” (AZ-1019-12), in whose context the presented study was conducted, is funded by the Bavarian Research Foundation (BFS).

The funder Josefinum, and private practice for Oral and Maxillofacial Surgery at Pferseepark provided support in the form of salaries for authors JW, but did not have any additional role in the study design, data collection and analysis, decision to publish, or preparation of the manuscript. The specific roles of these authors are articulated in the ‘author contributions’ section.

Within your Competing Interests Statement, please confirm that this commercial affiliation does not alter your adherence to all PLOS ONE policies on sharing data and materials by including the following statement: "This does not alter our adherence to PLOS ONE policies on sharing data and materials.” (as detailed online in our guide for authors http://journals.plos.org/plosone/s/competing-interests <about:blank> ) . If this adherence statement is not accurate and there are restrictions on sharing of data and/or materials, please state these. Please note that we cannot proceed with consideration of your article until this information has been declared.

Please know it is PLOS ONE policy for corresponding authors to declare, on behalf of all authors, all potential competing interests for the purposes of transparency. PLOS defines a competing interest as anything that interferes with, or could reasonably be perceived as interfering with, the full and objective presentation, peer review, editorial decision-making, or publication of research or non-research articles submitted to one of the journals. Competing interests can be financial or non-financial, professional, or personal. Competing interests can arise in relationship to an organization or another person. Please follow this link to our website for more details on competing interests: http://journals.plos.org/plosone/s/competing-interests <about:blank> 

Updated Competing Interests Statement:

The authors have declared that no competing interests exist.

The affiliation Josefinum, and private practice for Oral and Maxillofacial Surgery at Pferseepark of JW does not alter our adherence to PLOS ONE policies on sharing data and materials.

Additional Editor Comments: 

The authors propose a well-conducted study to compare topology optimized, patient specific osteosynthesis plates (TOPOS-implants) versus 1.0 mm miniplates by biomechanical testing using fibula-reconstructed cadaveric mandibles.

Introduction: To increase the clinical relevance, please include data about the benefits of PSI versus plates, referring to some clinical outcomes: recovery time for patients, shorten surgical and hospital time, etc.

A corresponding paragraph was added to the Introduction.

The aim of the study should be stated

A paragraph stating the aim of the study was added to the end of the Introduction.

The topology optimization of implants, the new technique tested in this manuscript, is defined as “…a powerful mathematical tool which allows creating an optimal structural design within prescribed loading and boundary conditions.”. This definition was supported by just one reference (#10) in page 4 line 76. This is a key point for this manuscript, so please add a more detailed definition and features of this specific technique to lead medical readers.

Thank you very much for pointing this out. More details about topology optimization and the advantages for osteosynthesis plates are added to the introduction.

Conclusions: They should state more precisely the benefits of TOPOS-implants over miniplates rather than “significant superiority”, which is an unprecise and subjective word.

The sentence was changed to:

This study displays a significantly increased cyclic loading capacity of mandible reconstructions with topology optimized and patient specific implants in comparison with miniplates as clinical standard

Reviewers' comments:

Reviewer's Responses to Questions

Comments to the Author

1. Is the manuscript technically sound, and do the data support the conclusions?

Reviewer #1: Yes

Reviewer #2: Yes

2. Has the statistical analysis been performed appropriately and rigorously? 

Reviewer #1: Yes

Reviewer #2: Yes

3. Have the authors made all data underlying the findings in their manuscript fully available?

The PLOS Data policy <http://www.plosone.org/static/policies.action#sharing> requires authors to make all data underlying the findings described in their manuscript fully available without restriction, with rare exception (please refer to the Data Availability Statement in the manuscript PDF file). The data should be provided as part of the manuscript or its supporting information, or deposited to a public repository. For example, in addition to summary statistics, the data points behind means, medians and variance measures should be available. If there are restrictions on publicly sharing data—e.g. participant privacy or use of data from a third party—those must be specified.

Reviewer #1: No

Reviewer #2: Yes

4. Is the manuscript presented in an intelligible fashion and written in standard English?

Reviewer #1: Yes

Reviewer #2: Yes

5. Review Comments to the Author

Reviewer #1: Abstract

Line 26: Please mention that “1.0mm” refers to thickness. Define “miniplates” and disclose their titanium grade.

Line 29: Thickness and other dimensions of the TOPOS plates should be mentioned.

Line 30: Titanium alloy? Grade?

The information about the materials of the used plates and the thickness of 1.0 mm was added to the abstract.

In Lines 212-215 you explain that the TOPOS plates fractured under lower static load than the miniplates. Also that the fibula fractured more quickly, but SD probably did not allow to reach statistical significance. Should this not be mentioned?

We agree that the information about the results from static testing would be interesting for the abstract. But due to the limited amount of words and in order to make the abstract more understandable, we reduced it to the sentence: “Static testing was used to confirm mechanical similarity between the reconstruction groups.”

Concerning the results of static testing there must be a misunderstanding, neither TOPOS-implant nor miniplates fractured. The reconstructions were tested in a whole and bone fracture was the main failure type, only once plastic deformation of miniplates occurred. A sentence referencing the result table for the static testing in the supplementary data was added to the manuscript. The differences between the mean values of the groups (elastic limit: 11 N, reconstruction failure: 13 N) are small compared to the SD and the absolute mean value for static testing. This is the reason why we concluded comparable mechanical properties in static testing for the two groups in the abstract.

The shape of the implant seems more important than any other characteristics (line 279). This can be mentioned in the abstract.

In general, the abstract can reflect the findings in a much better and much understandable way.

To further emphasize the importance of the shape of the implants, it was included in the introductory words of the abstract.

M&M

Line 135: is high stiffness (= high brittleness) correctly the goal? Would reduced plasticity and increased elasticity not be a better criterion? Please explain in your text.

Brittleness as a material property often comes together with a high material stiffness. But in this case, the optimization goal was to maintain a high stiffness for the whole implant despite significant material reduction. The Young’s moduli, which are a measure for elasticity and can be seen as the material stiffness, are very similar for titanium grade 4 and grade 5. So, the material properties are not changed and the implant still has a ductile mechanical behavior. The created geometry is optimized to allow as little deformation as possible with the given material. This is criterion is chosen to have a better distribution of internal stress in the implant and to improve fatigue properties. An optimization of the implant to higher elastic deformation would have been possible, but this always comes with a decreased stiffness for the implant. In consequence the reconstruction is not as stable and the risk of interfragmentary motion is higher, which impedes proper bone healing.

Line 164: what do you mean with “surgical procedures”?

Thank you for pointing that out. The sentence can be misleading to the reader, because the in-vitro reconstruction is not comparable with the intraoperative situation. The sentence was changed to: “The fixation of corresponding bone segments was performed with standard surgical equipment. The position of the screws was chosen for every reconstruction individual by surgeon.”

Line 165: why did you choose for the same (small diameter) screws? Luckily none failed. Is the position of the screws not important? The configuration should be explained and the influence of the latter on bone fracture.

The same diameter of screws was chosen to have comparable conditions for both reconstruction techniques. With only two length of screws the variational parameter were kept at a minimum, while maintaining the capability to create stable reconstructions. The position of the screws was chosen by the operating surgeon either at planning for TOPOS-implants or during reconstruction for the miniplates. That the position of the screws plays an important role is part of the results of this paper. This is especially important for the mandible angle, where a weakening effect of the screws becomes obvious. 

Line 176-182: this is difficult to understand. Please rephrase. Did you take laterotrusion (= horizontal shear forces at the occlusal level in the molar area) and protrusion (= vertical shear forces at the incisors) into account?

We did a restructuring of the paragraph with additional information added for better understanding. The force direction for the testing setup was derived from the major muscle groups. The in-vivo forces that are resulting from the muscular movement are dependent on the contact situation of the teeth. This includes shear forces at the incisors and the molar area. Due to the fact that we have only a bite point contact at the premolar, which is constraint in vertical direction for the setup, shear forces at the incisors are not present. Horizontal shear forces can occur for the setup at the bite point due to frictional contact. But this was not emphasized for this study. The loading situation was derived from different numerical and experimental models for testing of the mandible. The bearing at the end of the left mandible segment creates the biggest lever arm for the loading, which is like a worst-case scenario for possible chewing loads considering the remaining teeth.

Line 180: name of device, name of company, city, country 

Thank you very much for this hint. The test system has already been described as custom made in a previous paragraph. To avoid confusion for the reader, this has been pointed out again in the manuscript.

Lines 212-218 belong to the Results section.

The paragraph was moved to the Result section.

Line 230: could Young modulus not be determined per specimen after static and cyclic loading, to test differences between mandibles, being probably a confounding factor?

The information about the stiffness of the reconstruction (static: stiffness in elastic region, dynamic: actuator displacement) was added to the supplementary information and also mentioned in the Result and Discussion sections. Especially the displacement of the actuators shows that the both reconstruction types show very similar mechanical reaction to the applied load. This eliminates the stiffness of the reconstruction as confounding factor for this study.

Line 236: did you perform a Shapiro-Wilk test to conclude this?

Yes, due to the fact that the specimen number was too small for a D’Agostino & Pearson omnibus test, a Shapiro-Wilk normality test was performed. This indicated that the results of the dynamic testing were not normally distributed. (Shapiro-Wilk normality test: miniplates group p = 0.002; TOPOS-group p = 0.0008). This information was added to the manuscript.

Results

Line 253: I read “No TOPOS-implant failed for the applied cycles” and then I read in Line 254: “Failure is noted in the TOPOS implant group”, which is confusing.

Thank you very much for pointing that out. The second sentence was changed to “…, reconstruction failure (including bone fractures) is noted (…) in the TOPOS implant group…”.

Overall a good paper, but difficult to read for a surgeon. Its impact may increase of you make it more accessible.

Thank you very much for this hint. We hope the revision made it easier accessible for the readers.

Reviewer #2: According to the study design by the authors, the research work it is according to be in the right way by the literature available. Could be interesting in the future to use the same design of study to evaluate the properties of PSI vs reconstruction plates. The only thing that I miss was the blind selection of the groups of miniplates vs PSI.

Thank you very much. A comparison with reconstruction plates does sound like an interesting consecutive project. 

The specimen in this study were first separated in two groups with a similar distribution of the parameters sex and age. This was done so that the result could be considered independent of these factors. Otherwise age and sex could have been confounding factors.

---

## [Editor Report · Decision Letter 1]

6 May 2021

PONE-D-20-39788R1

Improving mandibular reconstruction by using topology optimization, patient specific design and additive manufacturing? – A biomechanical comparison against miniplates on human specimen

PLOS ONE

Dear Dr. LANG,

Thank you for submitting your manuscript to PLOS ONE. After careful consideration, we feel that it has merit but does not fully meet PLOS ONE’s publication criteria as it currently stands. Therefore, we invite you to submit a revised version of the manuscript that addresses the points raised during the review process.

Dear authors, thank you for submitting this revised version of the manuscript: "Improving mandibular reconstruction by using topology optimization, patient-specific

design and additive manufacturing? – A biomechanical comparison against mini plates

on human specimen" by Lang et al.

As a general comment, the revision of the final revised version (PONE-D-20-39788R1) was complex because it doesn't include changes highlighted in red or yellow color. All changes were presented at the end of the compiled PDF file document as raw tracked changes. TO MAKE THE NEXT REVISION EASY, PLEASE, BE SURE TO HIGHLIGHT FINAL MODIFICATIONS IN RED COLOR ONLY IN THE FINAL REVISED VERSION.  

Reviewer 1 has 2 comments/suggestions. Please, address them in the next 30 days.

We look forward to receiving your revised manuscript.

Kind regards,

Dr. Luis Cordova

Academic Editor

PLOS ONE

Journal Requirements:

Additional Editor Comments (if provided):

Dear authors, thank you for submitting this revised version of the manuscript: "Improving mandibular reconstruction by using topology optimization, patient-specific

design and additive manufacturing? – A biomechanical comparison against mini plates

on human specimen" by Lang et al.

As a general comment, the revision of the final revised version (PONE-D-20-39788R1) was complex because it doesn't include changes highlighted in red or yellow color. All changes were presented at the end of the compiled PDF file document as raw tracked changes. TO MAKE REVISION EASY, PLEASE, BE SURE TO HIGHLIGHT FINAL MODIFICATIONS IN RED COLOR IN THE THE FINAL REVISED VERSION.

Reviewer 1 has two comments/suggestions located at the end of the paragraphs copied below. Please, address them in the next 30 days.

Best regards

Prof. Luis Cordova

Line 135: is high stiffness (= high brittleness) correctly the goal? Would reduced

plasticity and increased elasticity not be a better criterion? Please explain in your text.

Brittleness as a material property often comes together with a high material stiffness.

But in this case, the optimization goal was to maintain a high stiffness for the whole

implant despite significant material reduction. The Young’s moduli, which are a

measure for elasticity and can be seen as the material stiffness, are very similar for

titanium grade 4 and grade 5. So, the material properties are not changed and the

implant still has a ductile mechanical behavior. The created geometry is optimized to

allow as little deformation as possible with the given material. This is criterion is chosen

to have a better distribution of internal stress in the implant and to improve fatigue

properties. An optimization of the implant to higher elastic deformation would have

been possible, but this always comes with a decreased stiffness for the implant. In

consequence the reconstruction is not as stable and the risk of interfragmentary motion

is higher, which impedes proper bone healing.

Reviewer 1: Then please explain the importance of stress shielding, the micro strains involved related to the Utah paradigm.

Line 180: name of device, name of company, city, country

Thank you very much for this hint. The test system has already been described as

custom made in a previous paragraph. To avoid confusion for the reader, this has been

pointed out again in the manuscript.

Reviewer 1: Difficult to confirm the results by a re-test by a third party. Please describe the flaw.
---

## [Author Response · Author response to Decision Letter 1]

21 May 2021

Dear Mr. Cordova,

dear Mr. Mommaerts,

dear Mr. Carrasco,

thank you very much für the second revision of our manuscript entitled “Improving mandibular reconstruction by using topology optimization, patient specific design and additive manufacturing? – A biomechanical comparison against miniplates on human specimen”. In the following I would like to respond to each point mentioned in your revision.

The reference list has been checked. All references are accessible and no retractions are present. During the two revision cycles the following references position have been added to the reference list: 10-18, 35-37

2. As a general comment, the revision of the final revised version (PONE-D-20-39788R1) was complex because it doesn't include changes highlighted in red or yellow color. All changes were presented at the end of the compiled PDF file document as raw tracked changes. TO MAKE THE NEXT REVISION EASY, PLEASE, BE SURE TO HIGHLIGHT FINAL MODIFICATIONS IN RED COLOR ONLY IN THE FINAL REVISED VERSION

I am sorry to hear about these inconveniences. I did use the normal Track changes function in Word, which includes a highlighted mark-up, as stated in the requirements for submission of revisions. Somehow, in the transition to the compiled pdf there must have been a problem with these markups. Nevertheless, in my pdf-summary of the upload the mark-up was tracked and highlighted normally. I did upload now tracked changes files as pdf in order to prevent this complication. If there is still a problem with the mark-up, please explain me, how exactly to solve this.

Comments by Reviewer 1

3. Reviewer 1: Line 135: is high stiffness (= high brittleness) correctly the goal? Would reduced plasticity and increased elasticity not be a better criterion? Please explain in your text.

Brittleness as a material property often comes together with a high material stiffness.

But in this case, the optimization goal was to maintain a high stiffness for the whole implant despite significant material reduction. The Young’s moduli, which are a measure for elasticity and can be seen as the material stiffness, are very similar for titanium grade 4 and grade 5. So, the material properties are not changed and the implant still has a ductile mechanical behavior. The created geometry is optimized to allow as little deformation as possible with the given material. This is criterion is chosen to have a better distribution of internal stress in the implant and to improve fatigue properties. An optimization of the implant to higher elastic deformation would have been possible, but this always comes with a decreased stiffness for the implant. In consequence the reconstruction is not as stable and the risk of interfragmentary motion is higher, which impedes proper bone healing.

Reviewer 1: Then please explain the importance of stress shielding, the micro strains involved related to the Utah paradigm..

Thank you very much for pointing this out. To emphasize for the reader, that stress shielding and bone loss due to reduced micro strains is not an issue for these implants, two sections were added to the manuscript. First, the optimization process was rephrased to show that an optimization goal for high stiffness does not result in an extraordinary stiffness for the implant. Secondly it was highlighted that the mechanical similarity of the reconstruction does also mean similar mechanical stresses for the fracture area.

The following sentence was added to the Material&Methods section for clarification:

“… In other words, the optimization is a material reduction process that results in the least possible loss of stiffness for the implant.”

The topic concerning stress shielding and micromotions in the fracture is now addressed with the following paragraph in the discussion section:

“The mechanical similarity also confirms that there is no excessive stiffening caused by the TOPOS implants. If the bone fragments are positioned too rigidly for osteosynthesis, it can result in inhibition of sufficient micromotions in the fracture gap [35,36]. Stress shielding like effects can lead to impaired healing in this context due to low interfragmentary strains at the fracture site [37]. But as already described there is no indication for this found in mechanical testing. “

4. Reviewer 1: Line 180: name of device, name of company, city, country 

Thank you very much for this hint. The test system has already been described as custom made in a previous paragraph. To avoid confusion for the reader, this has been pointed out again in the manuscript.

Reviewer 1: Difficult to confirm the results by a re-test by a third party. Please describe the flaw.

The following sentence was added to the manuscript for clarification:

“The biomechanical testing in this study was performed on an in-house developed testing setup. Using a custom-made solution like this comes with the disadvantage of limited reproducibility for third parties. But a commercially available solution does not exist for this testing scenario. And due to the detailed description of the testing and the publication presenting the experimental setup by Foehr et al. [26], distinct transparency, comprehensibility and adaptability of the experiments are ensured.”

We hope the revised version of our article meets all the conditions for publication in your journal.

---

## [Editor Report · Decision Letter 2]

27 May 2021

Improving mandibular reconstruction by using topology optimization, patient specific design and additive manufacturing? – A biomechanical comparison against miniplates on human specimen

PONE-D-20-39788R2

Dear Dr. Lang,

We’re pleased to inform you that your manuscript has been judged scientifically suitable for publication and will be formally accepted for publication once it meets all outstanding technical requirements.

Kind regards,

Dr. Luis Cordova

Academic Editor

PLOS ONE
---

## [Editor Report · Acceptance letter]

31 May 2021

PONE-D-20-39788R2 

Improving mandibular reconstruction by using topology optimization, patient specific design and additive manufacturing? – A biomechanical comparison against miniplates on human specimen 

Dear Dr. Lang:

I'm pleased to inform you that your manuscript has been deemed suitable for publication in PLOS ONE. Congratulations! Your manuscript is now with our production department. 

Kind regards, 

on behalf of

Dr. Luis Cordova 

Academic Editor

PLOS ONE